# Oral Glucocorticoid Use and Long-Term Mortality in Patients with Chronic Musculoskeletal Non-Cancer Pain: A Cross-Sectional Cohort Study

**DOI:** 10.3390/diagnostics13152521

**Published:** 2023-07-28

**Authors:** Hey-Ran Choi, In-Ae Song, Tak Kyu Oh

**Affiliations:** 1Department of Anesthesiology and Pain Medicine, Inje University Seoul Paik Hospital, Seoul 04551, Republic of Korea; viovio04@naver.com; 2Department of Anesthesiology and Pain Medicine, Seoul National University Bundang Hospital, Seongnam 13620, Republic of Korea; songoficu@outlook.kr; 3Department of Anesthesiology and Pain Medicine, College of Medicine, Seoul National University, Seoul 01811, Republic of Korea

**Keywords:** pain, glucocorticoids, steroids, arthritis, rheumatoid, osteoarthritis, gout

## Abstract

This study aimed to examine the associated factors of oral glucocorticoid (GC) use in patients with chronic non-cancer pain (CNCP) associated with musculoskeletal diseases (MSDs) in South Korea. Moreover, we examined whether oral GC use was associated with long-term mortality in patients with CNCP. This population-based cohort study used data from the national registration database in South Korea. Using a stratified random sampling technique, we extracted the data from 2.5% of adult patients diagnosed with MSDs in 2010. Patients with CNCP-associated MSDs who were prescribed oral GC regularly for ≥30 days were defined as GC users, while the other patients were considered to be non-GC users. A total of 1,804,019 patients with CNCP were included in the final analysis, and 9038 (0.5%) patients were GC users, while 1,794,981 (95.5%) patients were non-GC users. Some factors (old age, comorbid status, pain medication use, and MSD) were associated with GC use among patients with CNCP. Moreover, in the multivariable time-dependent Cox regression model, GC users showed a 1.45-fold higher 10-year all-cause mortality (hazard ratio: 1.45, 95% confidence interval: 1.36–1.54; *p* < 0.001) than non-GC users. In South Korea, the 10-year all-cause mortality risk increased in the patients with CNCP using GC.

## 1. Introduction

Chronic non-cancer pain (CNCP) is chronic pain that is not associated with cancer [1], and its global prevalence was reported to be one adult in five in 2010 [2]. CNCP is associated with increased substance abuse, emergency department utilization, and inpatient hospitalization [3], exerting a substantial social and economic burden on affected individuals and societies [4].

Glucocorticoids (GCs) are steroid hormones and are commonly prescribed among patients with chronic medical illnesses for 1.2% in the United States [5] and 1.0% in the United Kingdom [6], respectively. Both immunosuppressive and anti-inflammatory effects are the main mechanisms of GCs [7], which inhibit the production of the two main inflammatory products such as prostaglandins and leukotrienes [8]. Therefore, oral GCs have been prescribed to patients with various chronic medical conditions because of their immunosuppressive and anti-inflammatory effects [9,10,11,12]. Moreover, oral GC can be prescribed as an adjuvant analgesic to patients who experience severe pain [13]. Therefore, patients diagnosed with musculoskeletal diseases (MSDs) with CNCP may be prescribed oral GC as an adjuvant analgesic. Oral GC use reportedly causes various side effects, such as dysfunction of major organ systems including gastrointestinal, cardiovascular, endocrine, and neuropsychiatric dysfunction [14]. Moreover, increased long-term mortality was observed in oral GC use in the general adult population in previous studies [15,16]. However, the association between long-term mortality and oral GC use in patients with MSDs and CNCP has not been elucidated.

Thus, we aimed to examine whether oral GC use was associated with long-term mortality. We hypothesized that oral GC use was associated with increased long-term mortality in patients with CNCP.

## 2. Materials and Methods

### 2.1. Study Design and Ethical Statement

We assert that all procedures contributing to this work comply with the ethical standards of the relevant national and institutional committees on human experimentation and the Helsinki Declaration of 1975, as revised in 2008. All human patient procedures were approved by the Institutional Review Board (IRB) of the Seoul National University Bundang Hospital (IRB approval number: X-2105-685-901). The Ethics Committee of (IRB of Seoul National University Bundang Hospital: X-2105-685-901) waived the requirement for informed consent because of the study’s retrospective nature.

### 2.2. Data Source

The National Health Insurance Service (NHIS) database was used as a data source. The NHIS database includes information regarding all disease diagnoses and prescriptions for any drug or procedure because it is the sole public health insurance system in South Korea. The International Classification of Diseases and Related Health Issues 10th Revision (ICD-10) was used to register disease diagnoses in the NHIS database. The study protocol was approved by the NHIS Ethics Committee (NHIS approval number: NHIS-2021-1-615).

### 2.3. Study Population

The ICD-10 codes in Appendix A were used to define all the patients with CNCP. MSDs associated with CNCP include rheumatoid arthritis (RA), osteoarthritis (OA), low back pain, neck pain, and other MSDs. Adult patients with CNCP-associated MSDs were included in this study.

### 2.4. GC Exposure

The prescription data of oral GC were initially collected from 1 January 2010, to 31 December 2019, and followed up to 31 December 2019. Patients with CNCP who were prescribed oral GC (prednisolone, methylprednisolone, or dexamethasone) for ≥30 days were defined as GC users. Individuals not prescribed any GC or prescribed GC for <30 days were classified as non-GC users. Exposure was defined as a prescription ≥30 days of oral GC based on the year, and the prescription date was used to determine this. For example, patients with a prescription of ≥30 days of oral GC on 1 January 2011, were considered as GC users in 2011, while those with a prescription of ≥30 days of oral GC on 31 December 2012, were considered as GC users in 2012. Although, if an oral GC user discontinued GC for more than 30 days within the same year, the patient was considered as a GC user for that year because the patient was exposed to GC for ≥30 days during that year.

Since only prescription history was used to define a GC user, it was included in the study even if a patient was prescribed for 30 days on 31 December 2010, and died before 30 January 2010.

In South Korea, when a physician prescribes a certain drug, such as GC, the relevant primary diagnosis has to be registered in the NHIS database for the patient to receive financial coverage for treatment costs. This study included the oral GC prescription data with registered ICD-10 codes of MSDs as the primary diagnosis. Therefore, oral GC prescribed in this study might be associated with CNCP-associated MSDs.

### 2.5. Covariates

Data on age and sex were extracted as demographic information. Information on employment status, household income level, and place of residence was collected to determine the socioeconomic status of patients with CNCP because socioeconomic status could affect pharmacological treatment in patients with CNCP [17]. Information on household income level was included in the NHIS database to determine the insurance premiums of the South Korean population. However, individuals who are too poor to pay their insurance premiums or have difficulty supporting themselves financially are assigned to the Medical Aid Program, a government program developed to cover almost all medical expenses to help reduce the burden of medical costs on individuals. Except for the patients in the Medical Aid Program, all patients were divided into four groups using the quartile ratio. Residences of all patients were classified into two groups: urban areas (Seoul and six other metropolitan cities [Incheon, Daejeon, Gwangju, Daegu, Ulsan, and Busan]) and rural areas (all other areas). Data on the Charlson comorbidity index (CCI) and details of any underlying disabilities were collected to determine the physical comorbidity status of the patients because chronic pain conditions associated MSDs could co-occur with various physical comorbidities [18]. The CCI was calculated using the registered ICD-10 codes, as shown in Appendix A. As patients with CNCP are prescribed pain medication such as opioids, gabapentin or pregabalin, paracetamol, and nonsteroidal anti-inflammatory drugs (NSAIDs), information regarding pain medication (≥30 days) was also collected because history of other pain medication reflects pain severity of patients with CNCP.

### 2.6. Statistical Analyses

The clinicopathological characteristics between GC users and non-GC users are presented as numbers with percentages for categorical variables and mean values with standard deviation (SD) for continuous variables. *T*-tests and Chi-square tests were used to compare continuous and categorical variables among clinicopathological characteristics between GC users and non-GC users. We constructed a multivariable Cox regression model for GC use among patients with CNCP in 2010. All covariates were included in the model for multivariable adjustment, and results are presented as odds ratios (OR) with 95% confidence intervals (CI). The Hosmer–Lemeshow test was used to confirm that goodness of fit in the multivariable model was appropriate.

As exposure to GC varied between the GC and non-GC users in 2010 throughout the evaluation period (2011–2019), as shown in Appendix A, exposure to GC prescription was considered to be a time-dependent variable. All the other covariates were included in the time-dependent Cox regression model for multivariable adjustment, based on data in 2010 as time-fixed covariates. All-cause death from 1 January 2011 to 31 December 2019, was set as the event in the multivariable time-dependent Cox model.

In addition, we constructed 17 multivariable time-dependent Cox regression models for the 10-year disease-specific mortality. In these 17 models, all covariates were included for multivariable adjustment. As a sensitivity analysis, we made a multivariable time-dependent Cox regression model for 10-year all-cause mortality considering a lag-time period. In this sensitivity analysis, the evaluation period of 10-year all-cause mortality was 2012–2019, after excluding deaths in 2011, considering 2011 as a lag-time period. A variance inflation factor <2.0 was used to confirm no multicollinearity between variables in all models. The Cox regression results are presented as hazard ratios (HRs) with 95% CIs. All statistical analyses were performed using IBM SPSS (version 25.0; IBM SPSS Statistics for Windows, version 25.0, Armonk, NY, USA), and statistical significance was set at *p* < 0.05.

## 3. Results

### 3.1. Study Population

Initially, all patients were screened using the ICD-10 codes of MSDs from 2010 to 2019. The data of 800,000,000 cases of medical resource utilization with diagnoses of MSDs were extracted from the NHIS database. Owing to the large sample size, data from 2.5% of cases with adult patients (aged ≥20 years) were extracted using a stratified random sampling technique, considering age and sex as exclusive strata for sampling. The study population was drawn from each stratum, with sizes proportional to the strata of the overall 800,000,000 cases. Finally, 19,645,161 cases were sampled using a stratified random sampling method using SAS version 9.4 (SAS Institute, Cary, NC, USA) 17. Among these, 874,171 patients diagnosed with cancer as a comorbidity were excluded to focus on those diagnosed with CNCP. Next, 756 patients who underwent surgery during the study period were excluded because acute pain after surgery could be a source of bias in determining patients with CNCP. Therefore, 18,770,234 adult patients were screened. Among them, 1,808,043 patients with CNCP in 2010 were selected. Finally, after excluding 4024 patients with CNCP who died in 2010, 1,804,019 patients with CNCP were included as shown in Figure 1.

### 3.2. GC Users in Cohort 2010

Table 1 shows the results of the comparison of clinicopathological characteristics between GC users and non-GC users in patients with CNCP in 2010. A total of 9038 (0.5%) patients were GC users, while 1,794,981 (95.5%) patients were non-GC users. The mean ages of the GC user and non-GC users were 57.9 ± 15.9 (mean age ± SD) and 45.9 ± 16.0 years, respectively (*p* < 0.001). Table 2 shows the results of the multivariable logistic regression model for GC use in the 2010 cohort with CNCP. Old age (OR: 1.02, 95% CI: 1.02–1.02; *p* < 0.001), living in a rural area (OR: 1.05, 95% CI: 1.01–1.09; *p* = 0.037), increased CCI (OR: 1.17, 95% CI: 1.16–1.19; *p* < 0.001), opioid use (OR: 5.40, 95% CI: 4.64–6.29; *p* < 0.001), gabapentin or pregabalin use (OR: 1.82, 95% CI: 1.63–2.04; *p* < 0.001), and NSAID use (OR: 3.49, 95% CI: 2.80–4.35; *p* < 0.001) were associated with higher odds of GC use among patients with CNCP. In addition, among MSDs, underlying RA (OR: 6.82, 95% CI: 6.45–7.21; *p* < 0.001), OA (OR: 1.19, 95% CI: 1.13–1.26; *p* < 0.001), and gout (OR: 1.77, 95% CI: 1.63–1.94; *p* < 0.001) were associated with higher odds of GC use among patients with CNCP.

### 3.3. Survival Analyses

Table 3 shows the results of the multivariable time-dependent Cox regression model for 10-year all-cause mortality among cohorts with CNCP in 2010. GC users showed a 1.69-fold higher 10-year all-cause mortality (HR: 1.45, 95% CI: 1.36–1.54; *p* < 0.001) than non-GC users. All other HRs with 95% CIs in the multivariable time-dependent Cox regression model were presented in Appendix A. In a sensitivity analysis considering the lag-time period, GC users showed a 1.39-fold higher 10-year all-cause mortality (HR: 1.39, 95% CI: 1.35–1.43; *p* < 0.001) than non-GC users. Table 4 shows the results of the survival analyses of the 10-year disease-specific mortality. The GC users showed a higher 10-year mortality than non-GC users due to infection (HR: 2.00, 95% CI: 1.64–2.57; *p* < 0.001); cancer (HR: 1.18, 95% CI: 1.05–1.30; *p* = 0.005); blood disease (HR: 3.25, 95% CI: 1.90–5.02; *p* < 0.001); circulatory disease (HR: 1.30, 95% CI: 1.20–1.45; *p* < 0.001); respiratory disease (HR: 3.00, 95% CI: 2.75–3.30; *p* < 0.001); digestive disease (HR: 1.28, 95% CI: 1.05–1.51; *p* = 0.032); skin disease (HR: 3.40, 95% CI: 1.85–6.00; *p* < 0.001); MSD (HR: 8.50, 95% CI: 6.25–10.25; *p* < 0.001); genitourinary disease (HR: 1.58, 95% CI: 1.25–1.95; *p* < 0.001); abnormal clinical and laboratory findings (HR: 1.30, 95% CI: 1.13–1.50; *p* = 0.001); and injury, poisoning and certain other consequences of external causes (HR: 1.22, 95% CI: 1.05–1.41; *p* = 0.010).

### 3.4. Proportion of GC Use in Patients with CNCP from 2010 to 2019

Figure 2 shows the proportion of GC users among patients with CNCP from 2010 to 2019. In 2010, the proportion of GC users was 0.5% (9194/1,808,043), gradually increasing to 0.8% (14,566/1,892,913) in 2019. The overall proportion of GC users among patients with CNCP was 0.6% (119,834/18,770,234) during 2010–2019.

## 4. Discussion

This population-based cohort study showed that certain factors (old age, comorbid status, pain medication, and MSD) were identified as potential risk factors for GC use among patients with CNCP. Moreover, GC use was associated with an increased 10-year all-cause mortality among patients with CNCP. Our results suggest that GC use may be a potential risk factor for poor long-term survival among patients with CNCP. The proportion of GC use also increased from 2010 to 2019 in South Korea.

Some analgesics, such as opioids, gabapentin, pregabalin, and NSAIDs, were associated with GC use among patients with CNCP. A previous cohort study reported that NSAIDs and opioid use were associated with GC use among patients with RA [19]. Moreover, GCs are commonly prescribed with opioids and NSAIDs for patients with rheumatic diseases, such as RA, ankylosing spondylitis, and psoriatic arthritis [20], which has been reported to be associated with increased NSAIDs and opioid use with GC among these patients. GC is an adjuvant analgesic often prescribed with other analgesics, such as NSAIDs and opioids [13,21]. Patients with MSDs and CNCP might be prescribed other analgesics, such as NSAIDs and opioids, first, and GC might be prescribed as a secondary adjuvant analgesic for patients with severe CNCP. In addition to RA, underlying OA and gout were associated with increased GC use among patients with CNCP. GC is prescribed to alleviate inflammation, pain, and other symptoms in patients with OA and gout [22]. However, the benefit of GC administration in patients with gout remains controversial [23] and requires further research.

GC users have lower average CCI points than non-users (as depicted in Table 1), while higher CCI was associated with higher odds of GC use among patients with CNCP (as depicted in Table 2). The results in Table 2 are more important because all covariates were included in that multivariable model. There are two points that might explain these results. First, patients with severe pain are more likely to have other underlying medical conditions [18]. Second, patients with chronic pain are likely to visit medical institutions more frequently than those without pain, so there is a possibility that other diseases can be diagnosed more easily [24].

The relationship between GC exposure and increased mortality has been examined in patients with asthma and RA [25,26]. In South Korea, the number of GC users in the adult population has grown and is associated with an elevated risk of long-term mortality [16]. Long-term GC administration is known to cause serious adverse effects such as immunosuppression, which increases the risk of infection [27] and impairs immunotherapy outcomes [28,29]. In this study, the risk of 10-year infectious mortality was 2.08-fold higher in GC users than in non-GC users among patients with CNCP.

Interestingly, the 10-year mortality risk due to MSDs was highest at 7.04 HR in GC users compared to non-GC users among patients with CNCP. In this study, most MSD-related deaths were due to age-related osteoporosis with current pathological fractures (M80). As long-term GC exposure is known to cause osteoporotic fractures [30], GC users among patients with CNCP had a higher mortality risk due to osteoporotic fractures than non-GC users. Moreover, the 10-year mortality risk due to respiratory diseases was higher (HR: 3.14) in GC users than non-GC users among patients with CNCP. Most respiratory disease-related deaths occur because of pneumonia (J18) and chronic obstructive pulmonary disease with acute lower respiratory tract infection (J44). An increased risk of infection is a well-documented major side effect of long-term GC exposure owing to its immunosuppressive effect [31]. Moreover, a previous cohort study reported that lower respiratory tract infection was the most common complication in patients who received systemic GC [32], consistent with the present study’s findings.

There were several limitations in this study. As we extracted only 2.5% of patients as sampled using a stratified random sampling technique, there might be differences between the sampled patients with CNCP and total patients with CNCP in South Korea. Second, we could not evaluate and adjust the severity of MSD. For example, the duration and severity of pain in each MSD might influence the use of GC and lead to increased long-term mortality in GC users. Third, some important information, such as body mass index and lifestyle factors, were not used as covariates in this study because the NHIS database did not contain it. Fourth, as we used GC prescription data for determining its users, the actual compliance of GC among GC users was not evaluated in this study. Fifth, as we used registered ICD-10 codes of MSDs to define our study population, it was not definite that our population with MSDs had CNCP. Therefore, the results should be carefully interpreted. Lastly, there might be the possibility of confounding by indication. GC users might have more severe diseases, such as medical conditions, leading to poor long-term survival outcomes. Moreover, GC could also be a surrogate of high MSD activities, which themselves already contribute to poor outcomes. Therefore, the results should be interpreted carefully.

## 5. Conclusions

In conclusion, certain factors, such as old age, comorbid status, pain medicines (opioid, gabapentin or pregabalin, NSAIDs) use, and MSD (RA, OA, and gout) were potential risk factors for GC use among patients with CNCP. In addition, the 10-year all-cause mortality risk increased among the patients with CNCP using GC. Our results suggest that patients with CNCP who were prescribed GC were at high risk of poor long-term survival outcomes.

## Figures and Tables

**Figure 1 diagnostics-13-02521-f001:**
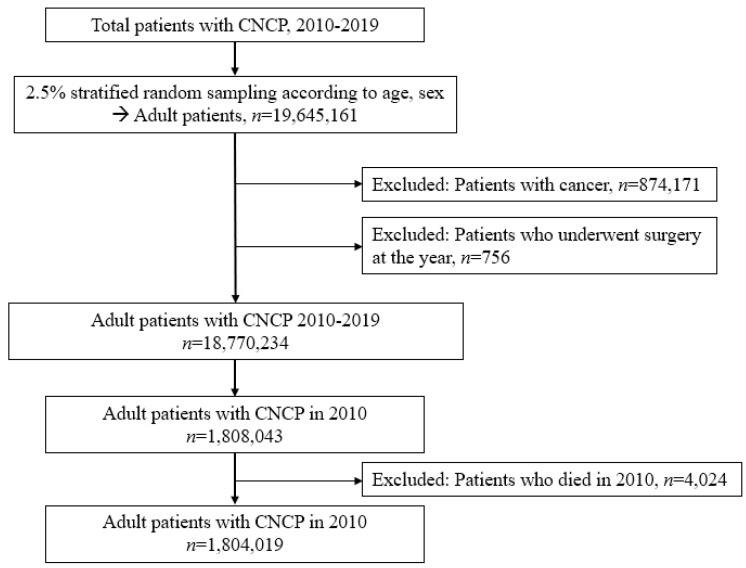
Patient selection flow chart.

**Figure 2 diagnostics-13-02521-f002:**
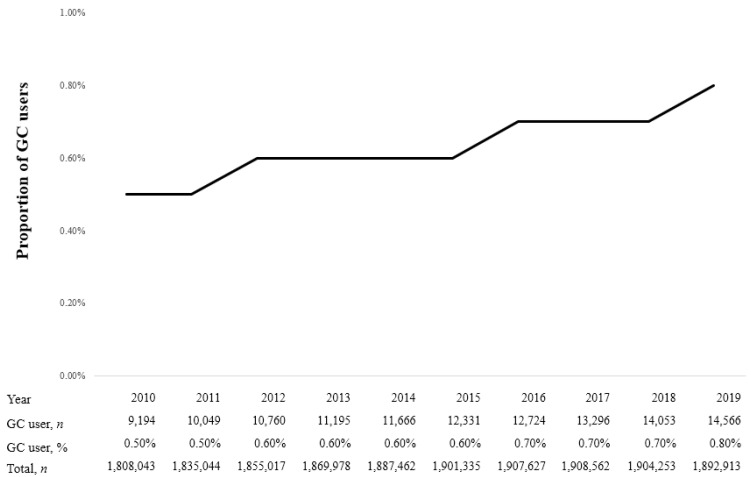
Proportion of glucocorticoid (GC) users in patients with chronic non-cancer pain (CNCP) from 2010 to 2019.

**Table 1 diagnostics-13-02521-t001:** Comparison of clinicopathological characteristics between GC users and non-GC users in patients with CNCP in 2010.

Variable		GC Users *n* = 9038	Non-GC Users *n* = 1,794,981	*p*-Value
Age, year		57.9 (15.9)	45.9 (16.0)	<0.001
Sex, male		3396 (37.6)	872,619 (48.6)	<0.001
Having a job		4435 (49.1)	954,005 (53.1)	<0.001
Household income				<0.001
	Medical aid	745 (8.2)	55,543 (3.1)	
	Q1 (lowest)	1573 (17.4)	332,888 (18.5)	
	Q2	1383 (15.3)	356,852 (19.9)	
	Q3	2083 (23.0)	446,160 (24.9)	
	Q4 (highest)	2951 (32.7)	546,506 (30.4)	
	Unknown	303 (3.4)	57,032 (3.2)	
Residence				<0.001
	Urban area	3839 (42.5)	834,290 (46.5)	
	Rural area	5199 (57.5)	960,691 (53.5)	
CCI		0.8 (1.2)	2.3 (1.9)	<0.001
Pain medication				
	Opioid use	733 (8.1)	7587 (0.4)	<0.001
	Gabapentin or pregabalin	414 (4.6)	8896 (0.5)	<0.001
	Paracetamol	531 (5.9)	6792 (0.4)	<0.001
	NSAIDs	102 (1.1)	1686 (0.1)	<0.001
Underlying MSDs				
	RA	2793 (30.9)	31,757 (1.8)	<0.001
	OA	4247 (47.0)	287,524 (16.0)	<0.001
	LBP	4362 (48.3)	422,588 (23.5)	<0.001
	Neck pain	1302 (14.4)	132,105 (7.4)	<0.001
	Gout	672 (7.4)	22,356 (1.2)	<0.001
	Other MSD	6326 (70.0)	449,647 (25.1)	<0.001

GC, glucocorticoid; CNCP, chronic non-cancer pain; CCI, Charlson comorbidity index; NSAIDs, non-steroidal anti-inflammatory drugs; MSD, musculoskeletal disease; RA, rheumatoid arthritis; OA, osteoarthritis; and LBP, low back pain.

**Table 2 diagnostics-13-02521-t002:** Multivariable logistic regression model for GC use in the 2010 cohort with CNCP.

Variable		OR (95% CI)	*p*-Value
Age, year		1.02 (1.02, 1.02)	<0.001
Sex, male (vs. female)		0.99 (0.95, 1.04)	0.668
Having a job		1.01 (0.96, 1.05)	0.771
Household income			
	Medical aid	1	
	Q1 (lowest)	0.85 (0.77, 0.94)	0.001
	Q2	0.78 (0.70, 0.86)	<0.001
	Q3	0.86 (0.78, 0.94)	0.001
	Q4 (highest)	0.85 (0.78, 0.93)	<0.001
	Unknown	0.89 (0.77, 1.02)	0.085
Residence			
	Urban area	1	
	Rural area	1.05 (1.01, 1.09)	0.037
CCI, point		1.17 (1.16, 1.19)	<0.001
Pain medication			
	Opioid use	5.40 (4.64, 6.29)	<0.001
	Gabapentin or pregabalin	1.82 (1.63, 2.04)	<0.001
	Paracetamol	0.80 (0.66, 0.94)	0.007
	NSAIDs	3.49 (2.80, 4.35)	<0.001
Underlying MSDs			
	RA	6.82 (6.45, 7.21)	<0.001
	OA	1.19 (1.13, 1.26)	<0.001
	LBP	0.96 (0.91, 1.01)	0.080
	Neck pain	0.92 (0.86, 0.98)	0.008
	Gout	1.77 (1.63, 1.94)	<0.001
	Other MSD	2.67 (2.53, 1.94)	<0.001

GC, glucocorticoid; CNCP, chronic non-cancer pain; OR, odds ratio; CI, confidence interval; CCI, Charlson comorbidity index; NSAIDs, non-steroidal anti-inflammatory drugs; MSD, musculoskeletal disease; RA, rheumatoid arthritis; OA, osteoarthritis; and LBP, low back pain.

**Table 3 diagnostics-13-02521-t003:** Multivariable time-dependent Cox regression model for 10-year all-cause mortality among cohort with CNCP in 2010.

Variable		HR (95% CI)	*p*-Value
Multivariable model			
	GC user (vs. non-GC users)	1.45 (1.36, 1.54)	<0.001
Sensitivity analysis considering lag-time period			
	GC user (vs. non-GC users)	1.39 (1.35, 1.43)	<0.001

GC, glucocorticoid; CNCP, chronic non-cancer pain; HR, hazard ratio; and CI, confidence interval.

**Table 4 diagnostics-13-02521-t004:** Survival analyses of the 10-year disease-specific mortality using time-dependent Cox regression model.

Mortality		Event (*n*, %)	HR (95% CI)	*p*-Value
Infectious mortality				
	Non-GC user	3091/1,794,981 (0.2)	1	
	GC user	81/9038 (0.9)	2.00 (1.64, 2.57)	<0.001
Cancer mortality				
	Non-GC user	28,271/1,794,981 (1.6)	1	
	GC user	307/9038 (3.4)	1.18 (1.05, 1.30)	0.005
Blood disease mortality				
	Non-GC user	281/1,794,981 (0.0)	1	
	GC user	11/9038 (0.1)	3.25 (1.90, 5.02)	<0.001
Endocrine disease mortality				
	Non-GC user	4527/1,794,981 (0.3)	1	
	GC user	89/9038 (1.0)	1.20 (0.90, 1.42)	0.072
Mental disease mortality				
	Non-GC user	2029/1,794,981 (0.1)	1	
	GC user	16/9038 (0.2)	0.78 (0.45, 1.25)	0.215
Nervous disease mortality				
	Non-GC user	4547/1,794,981 (0.3)	1	
	GC user	62/9038 (0.7)	1.21 (0.85, 1.42)	0.320
Circulatory disease mortality				
	Non-GC user	24,965/1,794,981 (1.4)	1	
	GC user	412/9038 (4.6)	1.30 (1.20, 1.45)	<0.001
Respiratory disease mortality				
	Non-GC user	11,881/1,794,981 (0.7)	1	
	GC user	450/9038 (5.0)	3.00 (2.75, 3.30)	<0.001
Digestive disease mortality				
	Non-GC user	4645/1,794,981 (0.3)	1	
	GC user	69/9038 (0.8)	1.28 (1.05, 1.51)	0.032
Skin disease mortality				
	Non-GC user	230/1,794,981 (0.0)	1	
	GC user	10/9038 (0.1)	3.40 (1.85, 6.00)	<0.001
Musculoskeletal disease mortality				
	Non-GC user	607/1,794,981 (0.0)	1	
	GC user	73/9038 (0.8)	8.50 (6.25, 10.25)	<0.001
Genitourinary disease mortality				
	Non-GC user	2876/1,794,981 (0.2)	1	
	GC user	74/9038 (0.8)	1.58 (1.25, 1.95)	<0.001
Mortality due to event during pregnancy, childbirth, and the puerperium				
	Non-GC user	15/1,794,981 (0.0)	1	
	GC user	0/9038 (0.0)	0.00 (0.00-)	0.995
Congenital disease mortality				
	Non-GC user	48/1,794,981 (0.0)	1	
	GC user	2/9038 (0.0)	3.48 (0.85, 15.85)	0.102
Mortality associated symptoms, signs, and abnormal clinical and laboratory findings				
	Non-GC user	10,300/1,794,981 (0.6)	1	
	GC user	149/9038 (1.6)	1.30 (1.13, 1.50)	0.001
Mortality due to injury, poisoning, and certain other consequences of external causes				
	Non-GC user	10,848/1,794,981 (0.6)	1	
	GC user	118/9038 (1.3)	1.22 (1.05, 1.41)	0.010
Mortality due to factors influencing health status and contact with health services				
	Non-GC user	41/1,794,981 (0.0)	1	
	GC user	0/9038 (0.0)	0.00 (0.00-)	0.970

GC, glucocorticoid; HR, hazard ratio; and CI.

## Data Availability

All data are available upon reasonable request from the corresponding author.

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
