# Peer review of "Oral Glucocorticoid Use and Long-Term Mortality in Patients with Chronic Musculoskeletal Non-Cancer Pain: A Cross-Sectional Cohort Study"

_diagnostics, 2023, doi:10.3390/diagnostics13152521_

Round 1
Reviewer 1 Report
The authors tried to examine whether oral glucocorticoid use was associated with long-term mortality in patients with chronic non-cancer pain. I appreciate the authors’ hard work. I have, however, several concerns.
Major concerns
#1. It is not clear why the authors selected the covariates (Page 3, Line 97).
#2. Why did the authors use Charlson Comorbidity Index (Page 3, Line 108)? Charlson Comorbidity Index seems out of date. I think using each comorbidity as a covariate is better (Ref. #15).
#3. GC users have lower CCI points than CCI nonusers (Table 1). Why higher CCI is associated with GC use (Table 2)?
#4. Tables 3&4 are missing.
Minor comments
#1. Please specify the organs (Page 2, Line 44).
#2. Please specify the cities (Page 3, Line108).
#3. I do not think that all patients with musculoskeletal diseases have chronic pain. How did the authors define patients with chronic pain (Page 2, Line 70)?
None
Author Response
The authors tried to examine whether oral glucocorticoid use was associated with long-term mortality in patients with chronic non-cancer pain. I appreciate the authors’ hard work. I have, however, several concerns.
Response: We appreciate your valuable comment and made every effort to revise our manuscript.
Major concerns
#1. It is not clear why the authors selected the covariates (Page 3, Line 97).
Response: We agree with your opinion, and the covariates section in the methods was revised to clarify our rationale for selecting the covariates with evidence from previous literature.
#2. Why did the authors use Charlson Comorbidity Index (Page 3, Line 108)? Charlson Comorbidity Index seems out of date. I think using each comorbidity as a covariate is better (Ref. #15).
Response: We agree that CCI might not be enough for reflecting comorbidity. However, it is widely used in most literature, and we did not use 17 for each comorbidity because there might be overfitting issues with including too many covariates in a multivariable model.
#3. GC users have lower CCI points than CCI nonusers (Table 1). Why higher CCI is associated with GC use (Table 2)?
Response: GC users have lower average CCI points than non-users (as depicted in Table 1), while higher CCI was associated with higher odds of GC use among patients with CNCP (as depicted in Table 2). The result in Table 2 is more important because all covariates were included in the multivariable model in Table 2. Two points might explain these results. First, patients with severe pain are more likely to have other underlying medical conditions (Davis et al., 2011). Second, patients with chronic pain are likely to visit medical institutions more frequently than those without pain, so there is a possibility that other diseases can be diagnosed more easily (Foley et al., 2021).
We revised the 3rd paragraph of the discussion section to clarify it.
#4. Tables 3&4 are missing.
Response: It was a mistake, and Tables 3 and 4 were inserted in the revised version of the manuscript.
Minor comments
#1. Please specify the organs (Page 2, Line 44).
Response: We corrected it according to your comment.
#2. Please specify the cities (Page 3, Line108).
Response: We corrected it according to your comment.
#3. I do not think that all patients with musculoskeletal diseases have chronic pain. How did the authors define patients with chronic pain (Page 2, Line 70)?
Response: As we used registered ICD-10 codes of MSDs to define our study population, it was not definite that our population with MSDs had CNCP. Therefore, the results should be carefully interpreted. We included this in the fifth limitation of the discussion section.

Reviewer 2 Report
The paper explored factors associated with mortality in chronic non-cancer MSK pain population.
Impression: this is a well-constructed paper with results that are likely of interest to clinicians. The following are addition suggestions that I believe should be considered to further improve the manuscript:
1.Please elaborate on how covariates were selected for inclusion into the regression model. Were they selected based on clinical reasoning, existing literature or was there any statistical methods used for covariate selection (e.g., univariable analysis)?
2.Please included valued of the regression coefficients in Table 2.
3.The discussion has mentioned many possibility that GC could contribute to mortality. However, I think GC could also be a surrogate of high MSD activities which themselves already contributed to poor outcomes. I suggest the authors include this point in the discussion.
no comment
Author Response
The paper explored factors associated with mortality in chronic non-cancer MSK pain population.
Impression: this is a well-constructed paper with results that are likely of interest to clinicians. The following are addition suggestions that I believe should be considered to further improve the manuscript:
Response: We appreciate your valuable comment and have made every effort to revise our manuscript.
1.Please elaborate on how covariates were selected for inclusion into the regression model. Were they selected based on clinical reasoning, existing literature or was there any statistical methods used for covariate selection (e.g., univariable analysis)?
Response: We agree with your opinion, and the covariates section in the methods was revised to clarify our rationale for selecting the covariates with evidence from previous literature.
2.Please included valued of the regression coefficients in Table 2.
Response: We did not include the regression coefficient in Table 2 because the odds ratio is familiar information for readers and easy to interpret. Moreover, a concise table might be more readable.
3.The discussion has mentioned many possibility that GC could contribute to mortality. However, I think GC could also be a surrogate of high MSD activities which themselves already contributed to poor outcomes. I suggest the authors include this point in the discussion.
Response: We agree with your opinion and have included the point in the last limitation of the discussion section.

Round 2
Reviewer 1 Report
The authors significantly improved manuscript. I have no further comments.